# Oligomer Sensor Nanoarchitectonics for “Turn-On” Fluorescence Detection of Cholesterol at the Nanomolar Level

**DOI:** 10.3390/molecules27092856

**Published:** 2022-04-30

**Authors:** Vedant Joshi, Sameer Hussain, Sachin Dua, Nishtha Arora, Sajjad Husain Mir, Gaulthier Rydzek, Thangaraj Senthilkumar

**Affiliations:** 1Polymeric Materials Area, Chemical and Material Sciences Division, CSIR-Indian Institute of Petroleum, Dehradun 248005, India; vedant.joshi@iip.res.in (V.J.); sachin.dua@iip.res.in (S.D.); nishtha.arora@iip.res.in (N.A.); 2Department of Chemical Science, Academy of Scientific and Innovative Research (AcSIR), Ghaziabad 201002, India; 3School of Chemistry, Xi’an Jiaotong University, Xi’an 710049, China; 4School of Chemistry and Advanced Materials & BioEngineering Research (AMBER) Center, Trinity College Dublin, The University of Dublin, D02 PN40 Dublin, Ireland; sajjad.mir@tcd.ie; 5ICGM, CNRS, ENSCM, University of Montpellier, 34000 Montpellier, France; gaulthier.rydzek@umontpellier.fr

**Keywords:** glycoconjugated oligomers, β-cyclodextrin appendage, host-guest complex, energy transfer, cholesterol detection, nanoarchitectonic sensors

## Abstract

Sensitive and rapid monitoring of cholesterol levels in the human body are highly desirable as they are directly related to the diagnosis of cardiovascular diseases. By using the nanoarchitectonic approach, a novel fluorescent conjugated oligofluorene (OFP-CD) functionalized with β-cyclodextrin (β-CD) was assembled for “Turn-On” fluorescence sensing of cholesterol. The appended β-CD units in OFP-CD enabled the forming of host-guest complexes with dabsyl chloride moieties in water, resulting in fluorescence quenching of the oligofluorene through intermolecular energy transfer. In the presence of cholesterol molecules, a more favorable host-guest complex with stoichiometry 1 cholesterol: 2 β-CD units was formed, replacing dabsyl chloride in β-CD’s cavities. This process resulted in fluorescence recovery of OFP-CD, owing to disruption of energy transfer. The potential of this nanoarchitectonic system for “Turn-On” sensing of cholesterol was extensively studied by fluorescence spectroscopy. The high selectivity of the sensor for cholesterol was demonstrated using biologically relevant interfering compounds, such as carbohydrates, amino acids, metal ions, and anions. The detection limit (LOD value) was as low as 68 nM, affirming the high sensitivity of the current system.

## 1. Introduction

Over the past decade, numerous researchers have sought to create functional systems composed of well-controlled nano-units [1,2]. This new concept, called nanoarchitectonics, has enabled the combination of nanotechnology with organic synthesis, materials science, biological and supramolecular approaches, resulting in nanosystems with well-defined structures and functions [3,4]. The resulting functional materials are of high relevance for the design and manufacture of sensors [5], based on non-covalent and supramolecular recognition of analytes that are detected using a range of outputs including optical, mechanical, and electrochemical [6,7,8].

Cholesterol is an important biomolecular precursor for the synthesis of vitamin D, bile pigments, and steroidal hormones in the animal body [9]. Although cholesterol is synthesized in the endoplasmic reticulum, most of the cholesterol is transported to the plasma membrane where its distribution is essential for maintaining cell integrity and other biological functions of animal cells [10]. The normal level of cholesterol found in human body fluid is 140–200 mg/100 mL (<5.2 mmol), and a high level (>5.2 mmol) of cholesterol leads to cardiovascular diseases and many other disorders [11]. Thus, the detection of cholesterol levels in body fluids is important to control various diseases and allow us to monitor cholesterol transport across the membrane. Various detection systems, such as chemical methods [12], enzymatic assays [13,14], analytical techniques (gas and liquid chromatography or mass spectrometry) [15,16], molecular imprinting approach [17], and fluorescence-based assays [18,19,20,21,22,23,24], were developed to monitor the cholesterol level in body fluids. For instance, the chemical method based on Abell-Kendall protocol relies on a multi-step procedure that involves saponification of cholesterol followed by extraction and final color development with acetic anhydride-sulfuric acid [12]. Concurrently, enzymatic assays involve the use of costly enzymes, although limits of detection (LOD) are usually low [13,14]. Recently, chromatographic and mass spectrometric methods are the most accurate and sensitive approaches for cholesterol detection; however, they require costly equipment and extensive sample pretreatment [15,16]. In the context of fluorometric detection of cholesterol, a thiazole-based chemosensor has been developed for monitoring cholesterol with an LOD of 26 µM [20]. Recently, fluorescent carbon dots prepared from β-cyclodextrin have also been explored for specific monitoring of cholesterol with an LOD of 0.20 µM [18]. In another approach, carboxymethyl β-cyclodextrin-grafted MOFs have been reported for the detection of cholesterol in human serum with an LOD as low as 0.4 µM [24]. Thus, there is a need for a simple, effective, and sensitive detection method/probe for monitoring cholesterol. In this context, an ideal sensor would *(i)* ensure molecular recognition of cholesterol molecules with *(ii)* high selectivity and sensitivity while *(iii)* providing a simple and rapid output, such as optical detection in the visible range. 

Cyclodextrins (CDs) are water-soluble macrocyclic molecules that are made up of α(1–4)-linked D-glucopyranose units [25]. These molecules exhibit a hollow hydrophobic cavity with the shape of a cone. Owing to such a distinct cavity, CD acts as a host molecule and is highly capable of encapsulating small organic molecules called guests. Generally, CDs exist in three forms, as α-CD, β-CD, and γ-CD, consisting 6, 7, and 8 D-glucose subunits, respectively [26]. Out of these three forms, β-cyclodextrin (β-CD) was identified as a prime building block for functionalizing nano-units to be integrated into nanoarchitectonic systems, [27,28,29] granting them selective adsorption and controlled release abilities towards organic molecules [30,31,32]. β-CD is widely known for its strong affinity towards cholesterol and for its use in extraction of cholesterol from body fluids, cell membranes, and cultured cells [33]. β-CD has a hydrophilic exterior, which provides water solubility and a hydrophobic interior utilized for specific recognition of hydrophobic guest molecules. Cholesterol tagged with fluorescent probes was reported for the analysis of cholesterol transport in animal cells, but they suffered from low LOD values [34]. Thus, the development of an alternative fluorescent probe based on β-CD to monitor cholesterol directly at very low levels is highly desirable.

Conjugated polymers (CPs) and oligomers are widely acclaimed for their excellent photophysical and electronic properties that allow researchers to explore their applicability in chemo and biosensors [35], fluorescence imaging [36], drug delivery [37], and disease treatment [38]. Although they possess a rigid backbone, they become water-soluble by grafting charged pendants and neutral polar units, which can also act as sites for selective recognition of analytes through various interactions. [39,40,41,42,43,44,45]. Hence, fluorescent probes based on CPs and oligomers are advantageous in solving the problem of low LOD, and providing amplified detection of analytes. Herein, we report utilization of glycoconjugated oligomer OFP-CD functionalized with β-CD for “Turn-On” fluorescence-based detection of cholesterol with a very low LOD value of 68 nM. OFP-CD is prepared and made water-soluble by appending β-CD units as pendant functional groups to the oligomer backbone. β-CD not only provides water solubility to the probe but also acts as a specific receptor site for cholesterol. Figure 1 represents the general principle of the sensor: oligofluorene derivative OFP-CD was covalently grafted with β-CD enabling host-guest recognition of molecules. When dabsyl chloride quencher (Q) is complexed, OFP-CD’s fluorescence is quenched initially due to energy transfer from OFP-CD to Q. Subsequent addition of cholesterol results in replacement of the quencher in β-CD cavities, ultimately enabling the “Turn-On” fluorescence response. The possibility of energy transfer, replacement of dabsyl chloride by cholesterol, and its host-guest complexation with β-CD, are verified from fluorescence sensing experiments, demonstrating both sensitivity and selectivity.

## 2. Materials and Methods

### 2.1. Materials

Hexylamine (CAS 111-26-2), triethylamine (CAS 121-44-8), dabsyl chloride (CAS 56512-49-3), cholesterol (CAS 57-88-5), bilirubin (CAS 635-65-4), glucose (CAS 50-99-7), sucrose (CAS 57-50-1), phosphate (CAS 4265-44-2), and creatinine (CAS 60-27-5) were purchased from Sigma Aldrich, Bengaluru, India. NaCl (CAS 7647-14-5), KCl (CAS 7447-40-7) and mono-6-*O*-(*p*-toluenesulfonyl)-β-cyclodextrin (CAS 67217-55-4) were purchased from TCI chemicals, Dehradun, India. Starting materials for the functional oligomer synthesis, including OFP-Br and β-cyclodextrin functionalized with a thiol group (β-CD-SH), were prepared according to the reported procedures [46,47]. Solvents were purchased from Merck, Darmstadt, Germany, and used without further purification. 

### 2.2. Measurements

^1^H-NMR spectrum was recorded on a Bruker Advance 500 MHz spectrometer. The molecular weight of the OFP-CD was analyzed using gel permeation chromatography (GPC) from Viscotek using THF as eluent. UV-Vis absorption spectra were measured on a Thermo Scientific Evolution (201) spectrophotometer equipped with quartz cuvettes. Fluorescence spectra were measured on a Hitachi F-4500 fluorometer equipped with a xenon lamp as the excitation source, and quartz fluorescence cuvettes were used for sample analysis with the proper excitation wavelength. JEM 2100 (JEOL, Tokyo, Japan) high-resolution transmission electron microscope (HR-TEM) was used to study morphology after drop-casting an aqueous solution of OFP-CD onto a copper grid at room temperature.

### 2.3. Synthesis of Glycoconjugated Oligomer OFP-CD

Fluorescent conjugated oligofluorene functionalized with β-cyclodextrin pendant groups (OFP-CD) was synthesized by using the thio-bromo click reaction according to literature [31]. OFP-Br (1 g, 1.77 mmol), β-CD-SH (4.06 g, 3.53 mmol), and triethylamine (0.5 mL, 3.53 mmol) were dissolved in 30 mL of acetonitrile and purged with nitrogen. Hexylamine (0.64 mL, 3.53 mmol), dissolved in 1 mL of acetonitrile, was added slowly into the above solution. After 1 h of reaction at room temperature, the solution was precipitated in cold acetone to recover the polymer. The functional oligomer was reprecipitated twice in acetone and dried under vacuum. The oligomer was purified by dialysis (membrane MWCO: 2000 Da) against water to remove any unreacted β-CD-SH molecules, and dried using a freeze drier. The final product was obtained as a powder with a 74% yield. OFP-CD was characterized by NMR and gel permeation chromatography (GPC) measurements. ^1^H-NMR (400 MHz, DMSO-*d_6_*) δ (ppm): 7.77 (dd, J = 68.7, 8.9 Hz, 10H), 5.90–5.67 (m, 29H), 4.85 (d, J = 21.5 Hz, 15H), 4.60–4.41 (m, 13H), 3.60 (dd, J = 31.6, 8.6 Hz, 61H), 2.00 (d, J = 8.0 Hz, 6H), 1.59–1.43 (m, 6H), 1.42 (d, J = 71.8 Hz, 4H), 1.05 (dd, J = 29.5, 6.8 Hz, 14H), 0.43 (s, 7H). The molecular weight (M_w_) from GPC was found to be 18200 Da, corresponding to an approximate polymerization degree of 6.5, with PDI = 2.7.

### 2.4. Preparation of Host-Guest Complexes of OFP-CD with Dabsyl Chloride and Quenching Studies

Host-guest complexation between OFP-CD and dabsyl chloride was studied by following the quenching of the OFP backbone’s fluorescence induced by energy transfer to complexed dabsyl chloride. OFP-CD (1 µM) and dabsyl chloride of various concentrations (from 0–10 µM) were mixed in water for 10 min at room temperature to form a stable host-guest complex. The fluorescence spectrum of each solution was recorded separately with an excitation wavelength of 365 nm.

### 2.5. Fluorescence Sensing of Cholesterol

The fluorescence sensing of cholesterol was performed in water by preparing a 1-milliliter solution of OFP-CD complexed with dabsyl chloride (OFP-CD = 1 µM; dabsyl quencher = 10 µM) that was kept in a quartz cuvette. Adequate quantities of cholesterol were then added to the cuvette to reach concentrations ranging from 0 µM to 20 µM. For each target concentration, the fluorescence spectrum was recorded using an excitation wavelength of 365 nm.

### 2.6. Determination of the Stoichiometry of Host-Guest Complexation Using Job’s Method

A series of solutions containing OFP-CD+Q and cholesterol were prepared while keeping the total concentration of components constant (100 μM). The mole fraction of the cholesterol in the solution was varied from 0 to 1, and the fluorescence spectra of the resulting mixtures (arising from fluorescence emission of OFP-CD) were measured using an excitation wavelength of 365 nm. In order to determine the stoichiometry of host-guest complexation between cholesterol and OFP-CD, the fluorescence intensity was plotted against the mole fraction of cholesterol in the solution, and the curve maximum was determined (Job’s plot).

### 2.7. Assessment of the Selectivity and Sensitivity of the Functional Oligomer Sensor

The selectivity of the nanoarchitectonic system towards cholesterol was studied by mixing various analytes, such as Na^+^, K^+^, glucose, cholesterol, sucrose, phosphate, bilirubin, and creatinine, at a concentration of 20 µM to the solution of OFP-CD+Q (OFP-CD = 1 µM; dabsyl chloride quencher = 10 µM) in water separately, and monitoring change in fluorescence spectra for each analyte under an excitation wavelength of 365 nm. In order to determine the probe’s sensitivity, the LOD value for cholesterol was determined by adding small cholesterol concentrations (ranging from 0 nM to 100 nM) to the aqueous solution of OFP-CD+Q (OFP-CD = 1 µM; dabsyl chloride quencher = 10 µM). The lowest concentration of cholesterol which induced a noticeable change in the fluorescence spectrum was considered as the lowest LOD for cholesterol [48]. Moreover, the limit of quantification (LOQ) was also calculated through multiplication of the LOD value by 3.3, according to reported method [49].

## 3. Results and Discussion

### 3.1. Design of OFP-CD as Functional Nano-Unit for Sensor Nanoarchitectonics

Grafting β-cyclodextrin moieties onto fluorescence-active OFP oligomers was performed to enable water solubility of the functional conjugated oligomer while granting it host-guest complexation abilities towards organic molecules containing hydrophobic domains. By applying the nanoarchitectonic approach, the combination of this functional oligomer unit with dabsyl chloride through host-guest complexation will enable quenching OFP’s fluorescence and thus completing the sensor. The synthesis approach of water-soluble glycoconjugated oligomer OFP-CD, described in Figure 1a, was performed following a two-step procedure: *(i)* first, the precursor brominated oligomer OFP-Br and thiolated β-cyclodextrin (β-CD-SH) were synthesized according to reported procedures [46,47,50], before *(ii)* being conjugated through thio-bromo click coupling to form OFP-CD with a 74% yield. The water solubility of the obtained functional oligomer is attributed to the grafting of hydrophilic cyclodextrin units onto the pendant chains, which compensate hydrophobicity of the oligomer backbone.

The final glycoconjugated oligomer OFP-CD was therefore soluble in most polar solvents, such as DMSO, DMF, MeOH, and water. Investigations by ^1^H-NMR analysis, along with labeling of chemical shifts (δ, ppm), confirmed the molecular structure of OFP-CD (Figure 1b). Peaks originating from protons in the fluorene and aromatic units were observed at δ 7.67–7.86 ppm, while protons from S-CH_2_ and O-CH_2_-S linkages were observed at δ 3.55 ppm, and δ 3.65 ppm, respectively. Finally, the protons arising from the backbone of β-CD units (labelled β-CD-OH on Figure 1b) were observed at δ 5.81, 4.88, and 4.54 ppm, respectively. Both the molecular weight and the polydispersity index of the oligomer OFP-CD were determined with GPC, using THF as eluent as 18200 Da with a PDI of 2.7. TEM images revealed that OFP-CD exhibit as spherical nanoparticles in water, with a size in the range of 150–200 nm, respectively (Figure 2). 

### 3.2. Fluorescence Quenching on Host-Guest Complexation of Dabsyl Chloride with OFP-CD

The functional oligomer OFP-CD was associated with dabsyl chloride (Q) via host-guest complexation in water (Figure 3a), and the quenching effect of dabsyl chloride was investigated by fluorescence measurements. Note that dabsyl chloride was chosen as a quencher due to its good solubility in water, ease of availability, and non-fluorescent nature. The normalized absorption (black line) and emission spectra (red line) of the OFP-CD in water (Figure 3b) clearly demonstrate a broad absorption maximum localized around 365 nm, and fluorescence emission maximum observed at 411 nm. On the contrary, dabsyl chloride did not exhibit any fluorescent behavior, yet its absorption spectrum indicated a strong absorbance around 460 nm (blue line on Figure 3b). Thus, the absorption spectrum of dabsyl chloride greatly overlaps with the emission spectrum of OFP-CD, inducing the possibility of efficient intermolecular energy transfer [39,43]. This possibility was investigated by studying the host-guest complexation of OFP-CD with increasing concentrations in dabsyl chloride from 0 to 10 µM, while monitoring the fluorescence emission spectra of the resulting mixtures. As shown on Figure 3c, the fluorescence intensity of the mixture decreased with increasing concentrations in dabsyl chloride, ultimately resulting in almost no emission for a concentration of 10 µM. This dramatic transition was also visible to the naked eye, as the color of the solution changed from blue to brownish when dabsyl chloride was added (Figure 3d). These results clearly indicate the quenching of OFP-CD’s fluorescence by dabsyl chloride in a concentration-related manner. In order to elucidate the quenching mechanism, the Stern–Volmer plot was acquired by following the evolution of I_0_/I emission intensities (where I_0_ is the initial fluorescence intensity of OFP-CD) as a function of the concentration of quencher (Q), as shown in Figure 3e. The non-linear nature of the obtained plot indicated that dabsyl chloride acts as a dynamic quencher, and absorbs all the fluorescence energy from OFP-CD after successful binding [51].

### 3.3. Cholesterol Detection by Quenched OFP-CD Nanoarchitectonic Sensor

The ability of dabsyl chloride to quench OFP-CD’s fluorescence on host-guest complexation with pendant β-cyclodextrin moieties was used to detect the presence of competitors for complexation in aqueous solution. Following this approach, the fluorescence sensing of cholesterol was carried out by monitoring changes in emission spectra of pre-quenched OFP-CD+Q complex ([OFP-CD]= 1 μM; [dabsyl chloride] = 10 µM) with various concentrations of cholesterol from 0 to 20 µM (Figure 4). On addition of cholesterol to pre-quenched OFP-CD solutions, the fluorescence emission spectra exhibited a peak of increasing intensity and corresponding to that of OFP-CD emission. Accordingly, the fluorescence color emitted by the solutions containing cholesterol under irradiation by a hand-held UV lamp was visible to the naked eye (inset in Figure 4a). This fluorescence recovery is attributed to the disruption of the energy transfer process between OFP-CD and the complexed quencher. This phenomenon is induced by competition between the dabsyl chloride quencher and cholesterol for host-guest complexation with β-CD moieties, ultimately resulting in replacement of the quencher by cholesterol in β-CD’s cavity. Note that the binding constant of cholesterol with cyclodextrin is one order of magnitude higher compared to that for the dabsyl-cyclodextrin complex [52].

To investigate the level of fluorescence recovery, the fluorescence intensity (F) observed after addition of cholesterol versus the initial fluorescence intensity (F_0_) was plotted against the concentration of cholesterol (Figure 4b). The plot signals a significant enhancement of fluorescence, with a maximum recovery exceeding 200 times upon addition of a 20-micromolar cholesterol concentration. These results confirmed that cholesterol largely replaced dabsyl chloride in β-CD’s cavities, resulting in the “Turn-On” fluorescence response of OFP-CD, with ~99% fluorescence recovery owing to disruption of the energy transfer process. In order to gain a deeper insight into the mode of complexation and stoichiometry for the OFP-CD/cholesterol host-guest complex, Job’s method was applied by changing the mole fraction of cholesterol in the aqueous mixtures, from 0 to 1, while observing the corresponding emission of OFP-CD. The fluorescence intensity of the functional oligomer versus the mole fraction of cholesterol exhibited an extremum at a 0.5 mole fraction of cholesterol; this indicated a stoichiometric ratio of cholesterol versus OFP-CD monomer of 1 (Figure 5). Since each OFP-CD monomer unit embeds two pendant chains, a single cholesterol molecule binds with two β-CD units of the fluorene backbone, which is consistent with the reported stoichiometry [52].

### 3.4. Practicability of OFP-CD-Based Nanoarchitectonic Sensor for Cholesterol Detection

To demonstrate the practicability of the nanoarchitectonic sensor, both the selectivity and the sensitivity of quenched OFP-CD towards cholesterol were investigated. First, the sensor’s selectivity was verified against a range of analytes (using a 20 µM concentration), which are biologically relevant, such as metal ions (Na^+^, K^+^), sugars (glucose, sucrose), anion (phosphate), creatinine, and bilirubin (Figure 6a). Figure 6a shows the selectivity of the probe in the presence of various other crucial interfering organic and inorganic agents that are usually found in human body fluids. For nearly all the tested analytes, the fluorescence response was at least an order of magnitude smaller than that induced by a similar concentration of cholesterol. Therefore, no significant interference was observed. In the case of bilirubin, a modest increment in the fluorescence of OFP-CD was measured, reaching an intensity of around 1.5 × 10^4^ compared to 10^5^ for cholesterol. This result signals hydrophobic interactions between the β-CD cavities and bilirubin molecules, enabling the replacement of some dabsyl chloride molecules in the host-guest complexes. However, this effect remained marginal in comparison to that of cholesterol. 

Finally, the sensor’s sensitivity was studied by spiking very low concentrations of cholesterol (~1–100 nM) with pre-quenched OFP-CD solutions to observe a noticeable increment in the fluorescence emission of OFP-CD. Interestingly, a very low LOD value of 68 nM and LOQ value of 224.4 nM were observed for cholesterol detection (Figure 6b), which outperforms several previous reports [18,19,20,21,22,23,24], as shown in Table 1. This clearly indicates the advantage of the nanoarchitectonic approach for assembling the current detection system: the combination of conjugated oligomers functionalized by pendant β-CD units enabled reversible fluorescence quenching through non-covalent host-guest complexation, which can act as a superior trigger for cholesterol sensing.

## 4. Conclusions

In conclusion, this study used the nanoarchitectonics approach to develop a glycoconjugated oligomer as a “Turn-On” fluorescent sensor for rapid, sensitive, and selective detection of cholesterol in water. First, a conjugated oligomer was functionalized with β-CD through a simple thio-bromo click reaction. The grafting of β-CD imparted good water solubility to the oligomer system, and also exhibited a strong affinity towards cholesterol. Next, the fluorescence of OFP-CD was reversibly quenched through non-covalent host-guest complexation with dabsyl chloride (quencher), enabling an intermolecular energy transfer process. Finally, the addition of cholesterol to solutions of pre-quenched OFP-CD successfully replaced the quencher from the cavities of β-CD, and subsequently facilitated the “Turn-On” fluorescence response. The selectivity of the system was confirmed using crucial interfering agents, such as various biomolecules, metal ions, anions, etc., that are commonly found in body fluids. The LOD value determined for cholesterol was as low as 68 nM using OFP-CD as the fluorescent probe, affirming the high sensitivity of the current system. The developed sensor exhibits likely futuristic applications for in vitro monitoring and imaging of cholesterol.

## Data Availability

The data presented in this study are available on request from the corresponding author.

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
