# Peer review of "Oligomer Sensor Nanoarchitectonics for “Turn-On” Fluorescence Detection of Cholesterol at the Nanomolar Level"

_molecules, 2022, doi:10.3390/molecules27092856_

Round 1

Reviewer 1 Report

In this manuscript, the authors developed a fluorescent sensor based on a fluorene-phenylene oligomer functionalized with cyclodextrin to detect cholesterol in biological conditions.  The fluorescence of the fluorene-phenylene oligomer was turned off by adding dabsyl chloride as a quencher (Q), which was induced into the cyclodextrin unit. The fluorescence was turned on again after the addition of cholesterol because the quencher was replaced from the cyclodextrin unit with cholesterol. In contrast, the fluorescence intensity was only slightly increased by adding other biologically relevant molecules. Thus, the high fluorene-phenylene oligomer showed high selectivity and sensitivity for detecting cholesterol. This manuscript is a very interesting development in the nanoarchitectonics and is suitable for Special Issue in the journal of Molecules. I recommend the publication of the paper after a minor revision. I would like to suggest the following modifications be made as necessary.  

1. I would like the authors to discuss about hos-guest chemistry, especially as cyclic molecules in Introduction. If the authors would explain the details of the cyclodextrin including the other types such as alpha and gamma, the manuscript will become better for wider audiences.  

2. What is the meaning of “PFP”? It was not defined in the manuscript. It seems that “PFP” means “polyーfluorene-phenylene”, however, the oligomer sensor has been developed as the title suggested. The authors should modify “PFP oligomer” to oligo-fluorene-phenylene.  

3. As shown in Figure 1, the authors described that the NMR peaks of δ 2.5 and 3.36 come from H2O and DMSO-d6, respectively. It is opposites, thus, DMSO shows NMR signal at δ 2.5 and solvent residual signals of water is at δ 3.33.  

4. Is it difficult to calculate j coupling constants and integral values of 1H from the NMR? It is difficult to get sharp peaks in polymer NMR, however, it seems that the peaks are enough sharp to calculate them, as shown in Figure 4. It means that it is better to indicate the replacement ratio of beta-cyclodextrin units from NMR. As shown in Figure 4, two beta-cyclodextrin units would be introduced into the polyfluorene backbone. 

Author Response

We would like to thank the reviewer for his valuable questions and suggestions. Please find our responses in the enclosed PDF file.

Reviewer 2 Report

This paper further expands the realms of nanoarchitectonics. The authors report fabrication of a sensing platform and demonstrate its feasibility and specificity for cholesterol detection. The authors synthesised and characterised fluorescent conjugated oligofluorene (PFP-CD) functionalised with β-cyclodextrin (β-CD). This paper is recommended for publication after some minor revisions:

  • the authors are suggested to comment on the applicability of the the sensor developed for microscopy imaging of cholesterol concentrations in vitro (e.g. in cell cultures)
  • can the authors provide some estimations on the size and morphology of the colloid clusters formed by these conjugated  cyclodextrins?
  • The authors should provide statistical evaluation results for plots in Figures 2,3 and 4
  • The authors list the following interfering molecules: "biomolecules, metal ions, sugar molecules, anions". Sugar molecules are biomolecules by definition. Please correct the terminology  

Author Response

(The authors gave the same response as above.)

Reviewer 3 Report

In this manuscript, Senthilkumar and co-workers are reporting the designing and application of an interesting nanoarchitectonics design for the detection of cholesterol by utilizing fluorescence spectroscopy. The manuscript introduces sufficient experimental evidence to support the application of the technique. Overall writing is also in a good format. One of the improvements I would like to suggest authors is to include more references for their explanations/claims within the introduction of the manuscript. This will improve the scientific quality and the reliability of the statements they are making. I would like to suggest following revisions prior to the acceptance of this manuscript.

(1). This probe is a blue light emitting device. It also requires high energy UV light for the excitations. Such light sources can be highly harmful in biological environments. Therefore, will this device be practically useful in biological applications? What is author’s opinion on this matter?

(2). Authors must provide a clear schematic representation for the inter-molecular energy transfer process and how it quenches the fluorescence of the probe.

(3). Authors are providing LOD values. But I do not see they explain how they obtain these values? Can authors possible describe or provide references to the method they used here? In addition, the most practical number in application is the limit of quantification (LOQ). Can authors also calculate the LOQ?

(4). In the introduction, authors should briefly explain other fluorescence-based analytical methods that has been developed so far for the detection of cholesterol and compare with the new method they have developed the highlight the specific advantages of this method.

(5). For all the detection methods that authors describing for cholesterol detection in the introduction, there is only one reference is give. Authors should provide all appropriate references to each method as well as the scope of the LODs for each technique.

(6). Authors are explaining the applications of CPs in the paragraph starting with line 75. Bioimaging with polymers are ot a common method. Due to the high molecular weight and the polymetric nature, these types of probes rarely get internalized in to live cells. This statement is misleading.

(7). What is authors opinion about using 10-times higher concentration of the quencher with respect to the probe? Does it mean the quenching mechanism is inefficient?

(8). Does, this probe sense any other polycyclic compounds such as naphthalenes, anthracenes or pyrenes?

(9). Is the quenching mechanism of the Dabsyl Chloride pH sensitive? Can authors provide experimental evidence?

(10). For figure 2e and 3b, I would suggest authors to use logarithmic plots to obtain the concentrations vs response plots.

(11). According to the Jobb’s plot, the highest fluorescence response obtains when the binding stoichiometry is 1:1. But this probe has ability to bind with 2 cholesterol moieties simultaneously. How does authors explain these observations? Does this mean when the fluorescence is turned ON, still there is one quencher molecule localized inside the probe?

(12). For figure 5a, I would also suggest adding the response upon addition of albumin as it is the common transporter protein in the biological environments.

Author Response

(The authors gave the same response as above.)
